# Humans can efficiently look for but not select multiple visual objects

Eduard Ort[1,2]*, Johannes Jacobus Fahrenfort[1,2,3], Tuomas ten Cate[4], Martin Eimer[5], Christian NL Olivers[1,2]

[1]Department of Experimental and Applied Psychology, Vrije Universiteit Amsterdam, Amsterdam, Netherlands; [2]Institute for Brain and Behavior Amsterdam, Amsterdam, Netherlands; [3]Department of Brain and Cognition, University of Amsterdam, Amsterdam, Netherlands; [4]Experimental Psychology, Utrecht University, Utrecht, Netherlands; [5]Department of Psychological Sciences, Birkbeck College, University of London, London, United Kingdom

**Abstract** The human brain recurrently prioritizes task-relevant over task-irrelevant visual information. A central question is whether multiple objects can be prioritized simultaneously. To answer this, we let observers search for two colored targets among distractors. Crucially, we independently varied the number of target colors that observers anticipated, and the number of target colors actually used to distinguish the targets in the display. This enabled us to dissociate the preparation of selection mechanisms from the actual engagement of such mechanisms. Multivariate classification of electroencephalographic activity allowed us to track selection of each target separately across time. The results revealed only small neural and behavioral costs associated with preparing for selecting two objects, but substantial costs when engaging in selection. Further analyses suggest this cost is the consequence of neural competition resulting in limited parallel processing, rather than a serial bottleneck. The findings bridge diverging theoretical perspectives on capacity limitations of feature-based attention.
DOI: https://doi.org/10.7554/eLife.49130.001

*For correspondence:
eduardxort@gmail.com

**Competing interests:** The authors declare that no competing interests exist.

## Introduction

Adaptive, goal-driven behavior demands the selection of relevant objects from the visual environment while irrelevant information is being ignored. This requires the neural activation of task-relevant representations in memory – often referred to as *attentional templates* – which then bias selection towards matching sensory input through top-down recurrent feedback loops (***Duncan and Humphreys, 1989***; ***Desimone and Duncan, 1995***; ***Hamker, 2004***; ***Eimer, 2014***; ***Cohen and Tong, 2015***). A fundamental yet unresolved question is whether the brain can enhance multiple task-relevant representations concurrently – a question that has recently generated considerable controversy, with arguments both for (***Houtkamp and Roelfsema, 2009***; ***Menneer et al., 2009***; ***Kristjánsson and Campana, 2010***; ***Dombrowe et al., 2011***; ***Olivers et al., 2011***; ***van Moorselaar et al., 2014***; ***Liu and Jigo, 2017***; ***Ort et al., 2017***; ***Ort et al., 2018***; ***van Driel et al., 2019***) and against (***Beck et al., 2012***; ***Irons et al., 2012***; ***Grubert and Eimer, 2015***; ***Grubert et al., 2016***; ***Beck and Hollingworth, 2017***; ***Kristjánsson and Kristjánsson, 2017***) a strong bottleneck.

We provide electrophysiological evidence showing that the real bottleneck is not so much in the number of different templates that can be concurrently active in anticipation of a visual task, but in the number of matching sensory representations in the incoming signal that can subsequently be enhanced by those templates. Crucially, for the selection of multiple targets to be truly simultaneous, two requirements have to be met. First, attentional templates need to be set up in memory for each anticipated target feature (whether in visual working memory [VWM] or through activating

long-term memory [LTM] representations). Although it is uncontroversial that multiple representations can be active in memory (whether VWM or LTM; e.g., *Cowan, 2001*), in order to be able to bias selection, each of these representations also needs to be in a state in which it can eventually engage, through recurrent feedback, with matching sensory signals (which is not the same as merely remembering; see *Carlisle et al., 2011*; *Olivers and Eimer, 2011*; *Kiyonaga et al., 2012*; *Chatham and Badre, 2015*; *van Driel et al., 2017*). Second, to simultaneously select multiple targets, the visual system must also be able to concurrently use those templates to strengthen multiple matching representations in the incoming sensory signal. In other words, multiple neural feedback loops must be able to engage concurrently. It is important to point out that template activation and template-guided prioritization are distinct (cf., *Huang and Pashler, 2007*): It may be that at any moment multiple templates are ready to potentially engage in the prioritization of visual input, but that only one can actually do so following visual stimulation. So far, studies of multiple-target selection have only focused on the brain's limits in the readiness to engage in selection, and ignored potential limits in the selection process itself.

To resolve this, we recorded electroencephalograms (EEG) from the scalp of healthy human individuals while they were presented with heterogeneous visual search displays, from which they always had to select two target objects (see *Figure 1A*). Crucially, we varied the number of unique target features (one or two colors) that the observer had to prepare for, and the number of unique features that they would need to select from the search display. This allowed us to disentangle the contribution of multiple template preparation on the one hand, and multiple template engagement on the other. A bottleneck could either emerge when going from one to two unique templates (reflecting a limit in the readiness to engage), from one to two unique targets (reflecting a limit in the engagement itself), or both.

Traditionally, visual target selection is assessed using the N2pc, an event-related potential (ERP) in the EEG signal that is characterized by increased negativity over posterior electrodes contralateral to the hemifield in which the target is located (*Luck and Hillyard, 1994*; *Eimer, 1996*). However, because the N2pc can only distinguish between the left versus right hemifield, it is not able to simultaneously track the selection of multiple targets at different locations in more complex visual search displays. To overcome this limitation, we used multivariate decoding, which has been proven to successfully track the spatiotemporal dynamics of feature-based selection processes at any location in a search display (*Fahrenfort et al., 2017*). Here this technique allowed us to independently track attentional selection over time for multiple concurrent targets at once, and also to investigate the parallel versus serial nature of these selection processes.

## Results

Twenty-four participants performed a visual search task for which they were always required to find two color-defined target characters presented among an array of distractor characters, and determine whether these two targets belonged to the same alphanumeric category (see *Figure 1A,B*). The task-relevant colors were cued prior to a block of trials. To assess if prioritization of multiple targets is limited in terms of the number of attentional templates that can be simultaneously set up, limited in the number of templates that can be simultaneously engaged in the selection of target features in the display, or both, we independently manipulated 1) how many colors were task-relevant and 2) how many of these target colors actually appeared in the search display. Specifically, in *1TMP–1TGT* (one template, one target feature) blocks, only one color was task-relevant, so that both targets had the same color and thus participants knew beforehand which color template to prepare. In *2TMP–1TGT* (two templates, one target feature) blocks, two unique colors were cued as task-relevant, but per display only one of these was used to distinguish the two targets present (i.e., both targets had the same color). Because participants could not predict which of the two target colors would be present, they had to keep both templates active, even though only one of them was then required for selecting the actual targets. Finally, in *2TMP–2TGT* (two templates, two target features) blocks, again two unique colors were cued as task-relevant, but now both these colors also had to be used to select the two target objects from the search display, since one of the targets carried one color, and the other target carried the other color. Note again that in all conditions, subjects had to select two targets, only the number of target-defining features would vary across

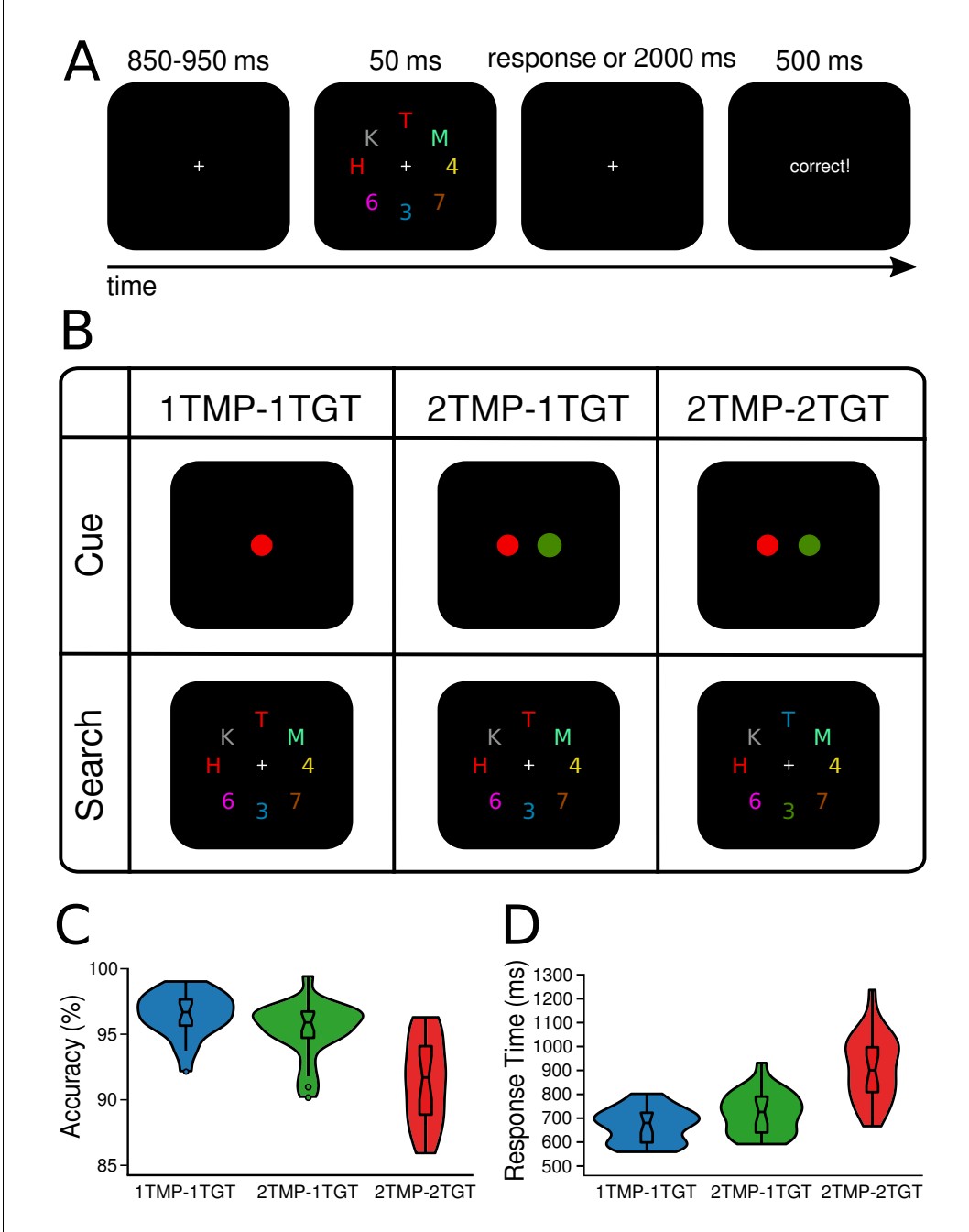

**Figure 1.** Design and behavioral results. In all conditions, observers were required to select two target characters and determine whether they were of the same (i.e., both letters or both digits) or different category (i.e., letter and digit). (**A**) The target colors were cued once in the beginning of a block and stayed constant for the rest of the block (64 trials). A trial started with a fixation screen for a jittered interval of 850 to 950 ms, followed by the search display, presented for 50 ms and another fixation screen that lasted for up to 2000 ms or until participants responded. Depending on the response, feedback was presented for 500 ms ('correct', or 'wrong'). If no response was given, a 10 s time out occurred and participants were urged to try responding quicker. (**B**) Task design. Depending on the condition, either one or two colors were cued to be task-relevant in the beginning of a block (creating one vs. two unique templates). Similarly, whenever two colors were cued, search displays could contain either one of them, or both (one vs. two unique target features). Thus, in the one-template-one-target-feature condition (*1TMP–1TGT*) one color was cued, and both targets carried this color in the search display, in the two-templates-one-target-feature condition (*2TMP–1TGT*) two colors were cued but only one of these colors was present in the search display with both targets carrying that color, and in the two-template-two-target-feature

*Figure 1 continued on next page*

*Figure 1 continued*

condition (*2TMP–2TGT*) two colors were cued and both colors were present in the search displays. One target always appeared on the horizontal meridian (above or below fixation), and the other target on the vertical meridian (to the left or right of fixation). (C) and (D) Behavioral results. The violin plots depict the distribution of (C) accuracy (see *Figure 1—source data 1*) and (D) response times (see *Figure 1—source data 2*) across participants, separately for the 1TMP–1TGT, 2TMP–1TGT, and 2TMP–2TGT conditions. The horizontal lines in the box plots represent quartiles. The vertical line represents the minimum (lower quartile - 1.5 * interquartile range) and maximum (upper quartile + 1.5 * interquartile range) while single dots beyond that range indicate individual outliers.

DOI: https://doi.org/10.7554/eLife.49130.002

The following source data is available for figure 1:

**Source data 1.** Accuracy data across participant that are represented in *Figure 1C*.
DOI: https://doi.org/10.7554/eLife.49130.003
**Source data 2.** Response time data across participant that are represented in *Figure 1D*.
DOI: https://doi.org/10.7554/eLife.49130.004

conditions. This controlled for other task-related factors such as the number of characters that had to be identified and the alphanumeric comparison that had to be performed on them.

## Behavioral results

*Figure 1C and 1D* show mean accuracy scores and mean response times (RTs) as a function of experimental condition (1TMP–1TGT, 2TMP–1TGT, and 2TMP–2TGT). Performance differences were assessed using pairwise, Bonferroni-corrected (to $\alpha$ = 0.025) classical *t*-tests and Bayesian *t*-tests on both measures. Any performance costs for the 2TMP–1TGT relative to the 1TMP–1TGT condition reflect the cost of preparing for multiple templates compared to a single template (*preparation cost*). Any performance cost in the 2TMP–2TGT relative to the 2TMP–1TGT condition represents the cost of having to engage multiple templates to select targets (*engagement cost*). We found evidence for both, with engagement costs being most prominent. Specifically, there was an effect of the number of templates on both accuracy and response times, with performance being reliably slower and slightly more error-prone in the 2TMP–1TGT condition than in the 1TMP–1TGT condition (RT: 731 ms vs. 679 ms, *t*(23) = 5.03, p<0.001, *Cohen's d* = 0.64, *BF* = 572; accuracy: 95.4% vs. 96.5%, *t*(23) = 2.76, *p* = 0.01, *Cohen's d* = 0.61, *BF* = 4.4). Even stronger costs were observed when the number of uniquely colored targets in the display was increased from one to two, with performance being substantially slower and more error-prone) in the 2TMP–2TGT condition than in the 2TMP–1TGT condition (RT: 916 ms vs. 731 ms, *t*(23) = 9.05, *p*<0.001, *Cohen's d* = 1.63, *BF* = $2.5 \times 10^6$; accuracy: 91.4% vs. 95.4%, *t*(23) = 5.90, p<0.001, *Cohen's d* = 1.48, *BF* = $3.9 \times 10^3$). Indeed, when we directly compared these two sources of multiple-target cost to each other, the engagement cost was greater than the preparation cost on both measures (accuracy: 4.0% vs. 1.2%, *t*(23) = 3.36, *p* = 0.03, *Cohen's d* = 1.03, *BF* = 14.8; RT: 185 ms vs. 52 ms, *t*(23) = 5.00, *p*<0.001, *Cohen's d* = 1.67, *BF* = 540).

Note further that in the 2TMP–1TGT condition, the actual target color in the display could repeat or switch from trial to trial. Previous work has shown switch costs, in which selection is slower after the target color changes from one trial to the next trial, compared to when the target color stays the same (*Maljkovic and Nakayama, 1994*; *Found and Müller, 1996*; *Monsell, 2003*; *Ort et al., 2017*; *Ort et al., 2018*). A closer analysis of the current data also revealed that search suffered from switches, in terms of RTs (repeat trials: M = 704 ms, switch trials: M = 754 ms; *t*(23) = 8.1, p<0.001, *Cohen's d* = 0.56, $BF_{switchcosts}$ = $4.2 \times 10^5$), and accuracy (repeat trials: M = 95.8%, switch trials: M = 94.9%; *t*(23) = 2.7, *p* = 0.01, *Cohen's d* = 0.40, $BF_{switchcosts}$ = 4.0).

The behavioral data thus reveal that multiple target search comes with costs, and that these costs come in two forms. First, keeping two templates in mind results in relatively small but reliable costs compared to keeping only one template. This effect is strongest when the actual target color in the display has switched, suggesting a shift in weights on specific templates from trial to trial. Second, considerably larger costs emerge when the observer not only maintains two different templates, but also has to engage both of them in biasing selection towards the two corresponding targets. Note that this is not the result of the number of target objects per se, as participants had to select and

compare two targets in all conditions, but it is caused by the number of unique features defining these targets. Selecting two objects by a single feature is thus more efficient and more accurate than selecting two objects using two different features.

## Decoding of target positions based on the raw EEG

Next, to determine whether the behavioral costs indeed reflected deficits in the selection of the different targets, we used EEG to track the strength and dynamics of attentional enhancement of the different target positions. To this end, one target was always placed on the vertical meridian, and the other target always on the horizontal meridian, so that we could train separate linear discriminant classifiers (with electrodes as features) for each of the spatial target dimensions to distinguish left from right targets and top from bottom targets, separately for each condition and time sample (see Materials and methods for details). We reasoned that any inefficiencies associated with setting up multiple unique templates (i.e., 1TMP vs. 2 TMP conditions) and/or with actually using those templates to select multiple unique targets (i.e., 1TGT vs. 2TGT conditions) should result in decoding to suffer in terms of relative delays, strength, or both. *Figure 2A* shows decoding performance for each of the conditions (1TMP–1TGT, 2TMP–1TGT, and 2TMP–2TGT), separately for the horizontal (left versus right) and vertical meridian (top versus bottom). *Figure 2B* shows the topographical patterns associated with the forward-transformed classifier weights over time, which are interpretable as neural sources (see *Haufe et al., 2014* and Materials and methods). As a general finding, we were able to track attentional selection on both the horizontal and vertical meridian, with comparable decoding performance. Decoding performance was tested against chance for every sample and corrected for multiple comparisons using cluster-based permutation testing (*Maris and Oostenveld, 2007*; also see Materials and methods). After cluster-based permutation, we observed clear significant clusters in each of the three conditions, with significant decoding emerging at different moments in time. For the left-right distinction, the topographical pattern during the early time window (200–350 ms) resembles that of the N2pc, while for later time windows (350–700 ms) it resembles SPCN or CDA-like patterns (*Vogel and Machizawa, 2004*; *Mazza et al., 2007*; *Grubert and Eimer, 2013*). As shown in *Figure 2—figure supplement 1*, more traditional event-related analyses indeed revealed N2pc and SPCN components, which likely contributed to the classifiers' performance. For vertically positioned targets a gradient from frontal to posterior channels spread along the midline, similar to recent results from our labs (*Fahrenfort et al., 2017*; *Grubert et al., 2017*). The fact that the decoding approach picks up on information related to attentional selection also on the vertical midline is testament to its power over conventional ERP methods, and allowed us to simultaneously track attentional selection of both targets over time. However, as there were no main or interaction effects involving the meridian in any of the comparisons, we averaged decoding performance across the spatial dimensions.

If there is a limit on how many templates can be prepared for, we should find reduced and/or delayed classification for the 2TMP–1TGT condition compared to the 1TMP–1TGT condition (*Figure 2C*). If the limitation is on how many templates can be engaged in selection, the cost should emerge in the comparison of the 2TMP–2TGT and 2TMP–1TGT conditions (*Figure 2D*). Indeed, we observed reliable differences for both comparisons that directly resembled the behavioral pattern. First, we compared the latencies at which target positions became decodable, thus providing a window on any delays in attentional selection. Because differences in onset of significant clusters cannot be reliably interpreted as reflecting differences in onsets of the underlying neurophysiological processes (*Sassenhagen and Draschkow, 2019*), we instead used a jackknife-based approach to quantify the latency of the 50% maximum amplitude in the decoding window (*Miller et al., 1998*; *Luck, 2014*; *Liesefeld, 2018*, see Materials and methods). This revealed a reliable onset difference between the 1TMP–1TGT (M = 216 ms) and 2TMP–1TGT (M = 237 ms) conditions (M = 21 ms, $t_c(23)$ = 2.21, $p$ = 0.04; *Figure 2C*), indicating that attentional selection is delayed as a result of having to prepare for two different target colors compared to having to prepare for only a single target color. Comparing the onsets between the 2TMP–1TGT (M = 237 ms) and 2TMP–2TGT (M = 263 ms) conditions yielded a further delay of 25 ms associated with having to engage in selecting two target colors compared to selecting a single target color ($t_c(23)$ = 2.35, $p$ = 0.03; *Figure 2D*).

Finally, and also similar to the behavioral responses, the onset of the neurophysiological response in the 2TMP–1TGT condition was delayed by 23 ms when the target color switched from one trial to

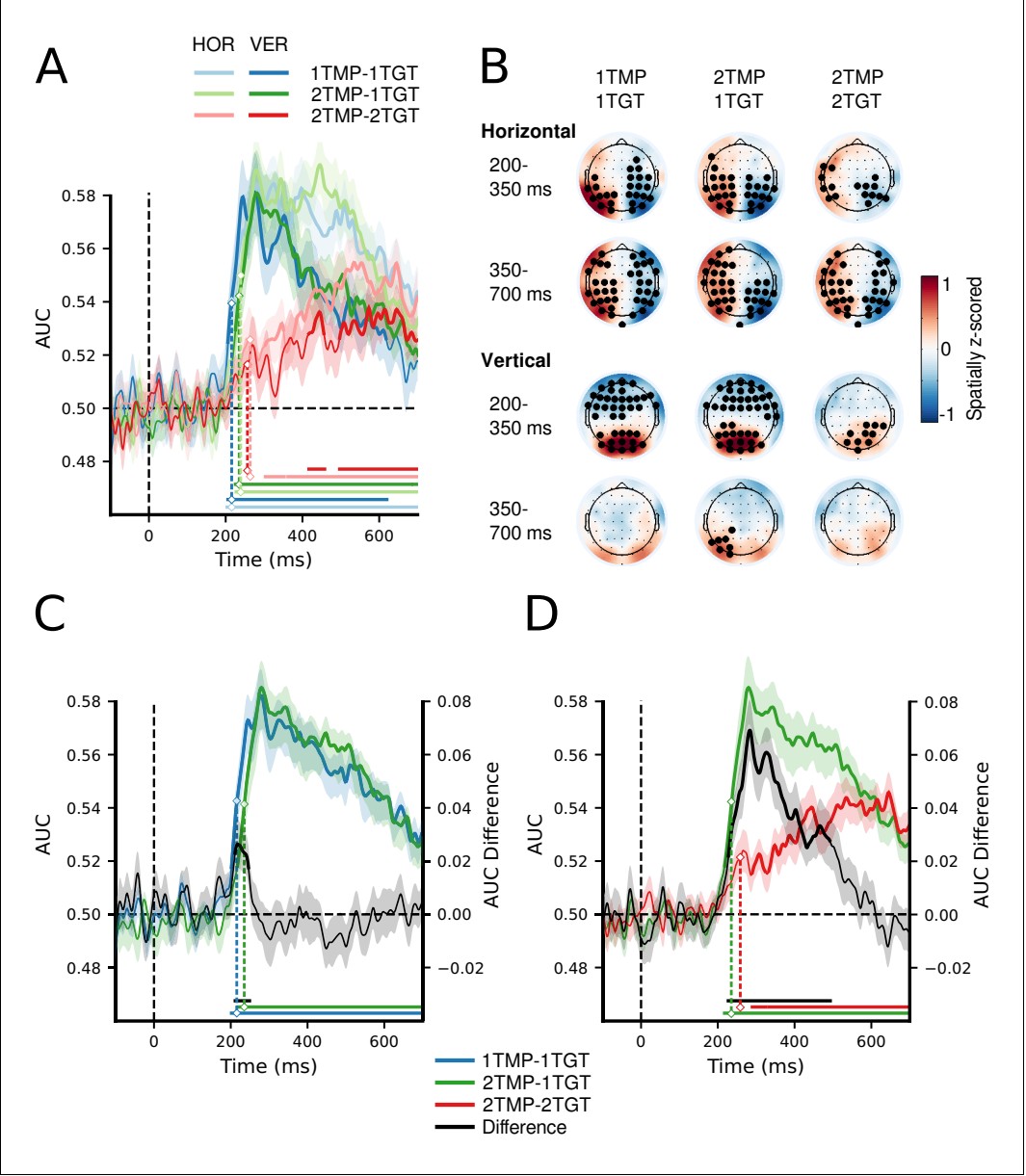

**Figure 2.** MVPA decoding performance for target position. (**A**) Decoding performance expressed as Area Under the Curve (AUC) for target position on the horizontal (left vs. right) and on the vertical meridian (top vs. bottom) separately, as a function of number of templates and number of target features. See also *Figure 2—source data 1* (**B**) Topographical activation maps for horizontal and vertical position decoding averaged over the typical N2pc time window (200–350 ms) and the typical SPCN/CDA time window (350–700 ms). (**C**) Decoding performance collapsed across the horizontal and vertical dimensions, comparing the 1TMP–1TGT and 2TMP–1TGT conditions, with the difference score thus showing the effect of the number of templates. See also *Figure 2—source data 2*. (**D**) The same, now comparing the 2TMP–1TGT and 2TMP–2TGT conditions, thus showing the effect of multiple different target features in the display. See also *Figure 2—source data 2*. The shaded area represents 1 SEM above and below the mean for every time point. Thick lines as well as horizontal bars indicate significant clusters (at $\alpha$ = 0.05) as produced by cluster-based permutation testing (5000 permutations). For visualization purposes only, the classification scores over time were fitted with a cubic spline ($\lambda$ = 15, comparable to a 30 Hz low-pass filter) to achieve temporal smoothing. Note the statistical analyses and estimation of the onset latencies were done on unsmoothed data. The marked time points indicate the latency of 50% maximum amplitude as estimated using a jackknife approach, as a measure of the onset of selection (*Miller et al., 1998*; *Luck, 2014*; *Liesefeld, 2018*). The zero points on the x-axis of panels A,C and D represent search display onsets.
DOI: https://doi.org/10.7554/eLife.49130.005

*Figure 2 continued on next page*

*Figure 2 continued*

The following source data and figure supplements are available for figure 2:

**Source data 1.** Time series data of classification AUC scores, separately for each condition and horizontal and vertical targets, as shown in *Figure 2A*.
DOI: https://doi.org/10.7554/eLife.49130.009

**Source data 2.** Time series data of classification AUC scores, separately for each condition (incl. separately for 21TGT switch and repeat trials), but collapsed across target position, as shown in *Figure 2C, D* and *Figure 2—figure supplement 2*.
DOI: https://doi.org/10.7554/eLife.49130.010

**Figure supplement 1.** Average N2pc difference waves for targets on the horizontal (left vs. right) meridian as a function of number of templates and number of target features.
DOI: https://doi.org/10.7554/eLife.49130.006

**Figure supplement 1—source data 1.** Time series data of N2pc difference wave locked to stimulus display onset for each condition separately.
DOI: https://doi.org/10.7554/eLife.49130.007

**Figure supplement 2.** MVPA decoding performance for target position separately for switch and repeat trials in the 2TMP-1TGT condition.
DOI: https://doi.org/10.7554/eLife.49130.008

the next, compared to when it repeated ($t_c(23) = 4.34$, p<0.001; see *Figure 2—figure supplement 2*).

Next, we assessed the strength of classification over time by testing AUC values of the relevant conditions against each other using paired *t*-tests and cluster-based permutation testing to correct for multiple comparisons (see Materials and methods). This procedure revealed an early and short-lasting difference of the number of templates (i.e., between 1TMP–1TGT and 2TMP–1TGT conditions; see *Figure 2C*), with stronger classification for the single template condition that reflects the onset latency difference reported above. Again, in line with the behavioral results, more substantial cost in decoding performance emerged when the number of target features in the displays increased from one to two (i.e., between the 2TMP–2TGT and 2TMP–1TGT conditions; see *Figure 2D*). To directly compare the cost of preparing two templates to the cost of engaging them in selection, we also ran a cluster-based permutation test on the difference scores (i.e. [2TMP–1TGT – 2TMP–2TGT] – [1TMP–1TGT – 2TMP–1TGT]). This revealed a window of 250 to 500 ms post stimulus in which the cost of engaging was greater than the cost of preparing selection (first cluster: extent: 266–378 ms, *p* = 0.001; second cluster: extent: 436–495 ms, *p* = 0.013, results not shown in Figure). This suggests that generally engaging two templates is more costly than preparing two templates.

Thus, both the onset latency and strength of decoding performance show clear deficits in attentional selection when observers need to select two different targets from a display (i.e., engage two templates in selection) compared to when they have to select two targets based on the same target color (i.e., engage one template in selection). In contrast, having to set up two templates instead of one came with only minor onset latency differences and no overall differences in decoding strength. This clearly points to a deficit when multiple templates need to be engaged simultaneously rather than when multiple templates need to be prepared simultaneously.

## Sample-wise correlation of classifier confidence across trials as a measure of inter-target dependency

While the previous section showed a clear impairment when two templates need to be engaged in selection, it leaves unanswered whether selection is hindered by limitations in parallel processing or by a serial bottleneck. That is, engaging two templates during search may prioritize both unique targets in parallel but in a mutually competitive manner (*Barrett and Zobay, 2014*), or the two templates may only be engaged (and thus the corresponding targets prioritized) sequentially, possibly in continuously alternating fashion (e.g., *Ort et al., 2017*).

To investigate these competing hypotheses, we assessed performance for each target dimension (horizontal and vertical) separately . A serial model predicts that attention to a target on one dimension should go at the expense of attention to the target on the other dimension, and thus decoding

performance for the vertical and horizontal axes to correlate negatively. In case of parallel, independent selection, there should be no systematic relationship between classification confidence for one dimension and classification confidence for the other dimension, as selection of one target is impervious to the selection of the other target. A positive correlation would arise from a common mechanism driving selection of two different targets. Note that these possibilities are difficult to assess at the group level as individuals may have different serial strategies. For example, one observer may prefer to first select targets from the horizontal axis, while another may prefer the vertical axis first, such that any existing correlation (if present) might cancel out. Hence, we first plotted average performance over time separately for each individual and separately for the horizontal and vertical axis. Then, to reveal whether consistent temporal dependencies existed for any given participant, we correlated classification performance over time in the 150 ms to 700 ms post stimulus window. Although this revealed incidental positive and negative correlations for individual participants, there was no systematically positive or negative relationship (average correlation *Spearman's* $\rho$ = 0.11; min-max range: −0.37–0.63; see *Figure 3—figure supplement 1*).

However, even individual participants themselves may not behave consistently across trials, and, while selection is still serial, whether participants first prioritize horizontal or vertical targets may also vary from trial to trial. Therefore, selection needs to be assessed at the trial level. To this end, for every participant, trial and time point, we extracted the classifier confidence scores separately for the horizontal and vertical dimension (see *Ritchie and Carlson, 2016*; *Grootswagers et al., 2018* and Materials and methods), and correlated the two dimensions across trials using *Spearman's* $\rho$. Classifier confidence, expressed as the distance from the decision boundary, reflects the certainty of a classifier in predicting the class membership of a certain instance. In the present design, classifiers predicted based on the specific EEG activity pattern across electrodes on a given trial whether one target appeared on the left or right and whether the other target appeared at the top or bottom position. The confidence scores indicate how certain the classifiers were that a target appeared at a particular position. We reasoned that if prioritization is limited to a single target at a time, a classifier cannot simultaneously have high confidence about both targets, and thus confidence should correlate negatively, that is if the horizontal target position can be predicted with high confidence, then the confidence for the vertical target position should be reduced and vice versa. The correlations between the confidence scores on the two spatial dimensions are plotted in *Figure 3A*. As can be seen, there was again no systematic relationship between decoding the locations of the two targets, in any of the conditions. Apart from a short-lasting positive correlation around the 500 ms time point in the 2TMP–2TG condition, which is likely to be spurious, correlations for all time points were close to zero, which implies that classifying horizontal and vertical target positions is independent from each other.

However, given that this is a null result, we sought to make sure that our approach is in principle sensitive to existing correlations. To this end, we simulated a data set with the same overall characteristics as the recorded data, but with either positive, negative, or no correlations injected, under various signal to noise ratios (see Materials and methods). The results of this simulation are summarized in *Figure 3—figure supplement 2* and demonstrate that with sufficiently high decoding AUC values (>approx. 0.55–0.60), correlations (whether positive or negative) between the horizontal and vertical position classifiers can, in principle, be reliably detected. However, because group classification performance in our dataset did not exceed 0.59 (in the 1TMP–1TGT condition), we instead assessed for each individual observer the correlation between target dimensions for those time points at which classification performance reached its maximum. As *Figure 3B* shows, even for individuals with relatively high classification scores, there was no evidence for a correlation between the classification confidence between the two target dimensions. The absence of such a correlation in our data is thus most consistent with a limited parallel independent selection model, rather than a serial model or a parallel model operating under a common mechanism.

Nevertheless, there is the possibility that there was actually a relationship between horizontal and vertical classification, but across time, trials, or both it was too short and inconsistent such that the present approach might not have been sensitive enough to detect it. For example, a negative correlation might exist for a short time window of which the timing shifted across trials, causing the resulting average correlation to be reduced beyond detectability. We emphasize that the conclusion that our data are mostly consistent with a limited parallel model is thus based on a null result and therefore has to be interpreted with care.

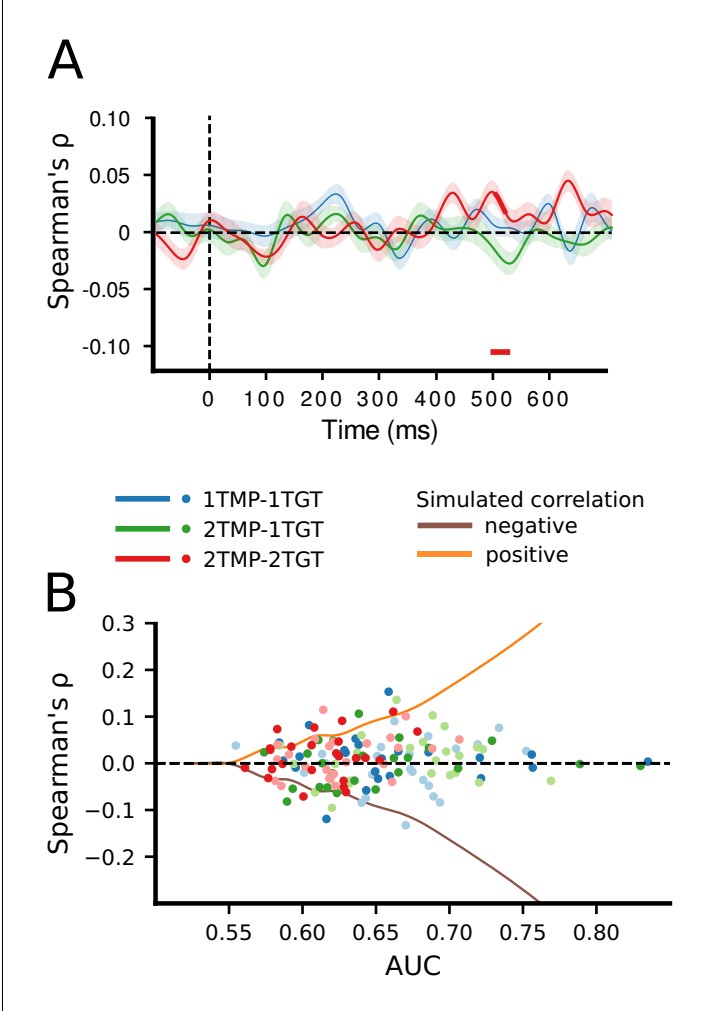

**Figure 3.** Correlation of classifier confidence scores. (**A**) For each condition, and trial, the classification confidence scores per time point and subject were extracted for horizontal and vertical classifiers and then correlated (*Spearman's ρ*) between these dimensions, across trials. (**B**) To examine whether a non-zero correlation would be present for individuals who show high AUC scores, we plotted the individual correlation scores for those time points at which individual classification was maximal, separately for the horizontal dimension (less saturated dots) and the vertical dimension (more saturated dots). The curves represent the correlation strengths that can be expected for a certain decoding strength (AUC, corresponding to SNR) as based on our simulated data set (the simulated negative correlation being the mirrored version of the positive correlation).

DOI: https://doi.org/10.7554/eLife.49130.011

The following source data and figure supplements are available for figure 3:

**Figure supplement 1.** Classification performance in the 2TMP-2TGT condition, separately for horizontal and vertical targets for each participant and individual correlation scores of classifier confidence scores between vertical and horizontal targets.

DOI: https://doi.org/10.7554/eLife.49130.012

**Figure supplement 1—source data 1.** Time series data of classifier confidence for horizontal targets in the 2TMP-2TGT condition per participant, as shown in *Figure 3—figure supplement 1*.

DOI: https://doi.org/10.7554/eLife.49130.013

**Figure supplement 1—source data 2.** Time series data of classifier confidence for vertical targets in the 2TMP-2TGT condition per participant, as shown in *Figure 3—figure supplement 1*.

DOI: https://doi.org/10.7554/eLife.49130.014

**Figure supplement 2.** Results of location decoding and correlation analysis of a simulated dataset across several signal-to-noise ratios (SNRs).

DOI: https://doi.org/10.7554/eLife.49130.015

# Discussion

Selection of task-relevant information from complex visual environments is limited, and a central question in attention research has been whether observers can simultaneously prepare for and select multiple different target objects. The current results provide evidence that these limitations do not so much reside at the level of template preparation (i.e., the number of target representations set up prior to the task), but at the extent to which templates can then be concurrently engaged in selecting matching information from the sensory input. By systematically varying not only the number of different target features observers had to prepare for, but also the number of different target features they would encounter in the displays, we were able to, for the first time, dissociate limitations in template preparation from limitations in template engagement. Specifically, we observed relatively small but reliable costs on both behavioral and EEG classification performance when two templates needed to be activated instead of one, suggesting a reliable but relatively minor bottleneck at this stage of processing. In contrast, substantial costs emerged on both behavioral and EEG performance measures when two templates had to be prepared, and both of these templates (rather than just one) had to be engaged in driving the selection of two different targets.

We propose a model which extends existing frameworks that assume a crucial role for top-down biased competition (*Duncan and Humphreys, 1989*; *Desimone and Duncan, 1995*; *Hamker, 2004*; *Bundesen et al., 2005*). According to these frameworks, the activation of target templates in memory involves the pre-activation or biasing of associated sensory features. The presence of such features in the input will then trigger a long-range recurrent feedback loop, leading the enhancement of the target representation in VWM (including its location), and thus making it available for other cognitive processes such as response selection (processes which are themselves limited, cf. *Dehaene et al., 1998*; *Lamme, 2003*; *Baars, 2005*). Our data indicates that while multiple top-down feedback connections may be prepared at once, there is a limitation in how these feedback loops are engaged by matching input.

*Figure 4* illustrates how we believe the existing framework should be extended. Specifically, we propose that multiple templates may hold each other in a mutually competitive relationship in memory, most likely through laterally suppressive connections (*Manohar et al., 2019*). *Figure 4A* depicts the situation when just one of the target features is then encountered in the sensory input. The corresponding feedback loop is triggered, leading to an enhanced representation of that target. If only one target feature is present, the corresponding template will automatically win the competition. Although two templates can be maintained in parallel, the mutual competition between them is slightly disadvantageous. This will lead to the initial delay in target selection that we observed in the data when two templates instead of just one were activated. Moreover, the selective enhancement of one representation over another may carry over to the next trial, thus resulting in the target switch costs that we also observed both in behavior and EEG performance measures.

The crucial situation occurs when the visual input contains multiple target features and thus multiple feedback loops are being triggered, as is shown in *Figure 4B*. Because of the mutually suppressive relationship, strengthening one feedback loop will automatically go at the expense of the other. Although both loops are triggered in parallel, the mutually aversive relationship results in slower and weaker accumulation of evidence for either of the targets, consistent with what we observed in the data. In theory, the system may resolve such competition in two ways. The first is to keep selection of both targets running in parallel, and accept the slower evidence accumulation. The second option is to impose a serial strategy in which selection is first biased in favor of one target, and then switched to the other (or alternate between the two). Our data provides no evidence for the serial model. First, the average group data nor the average individual subject data showed any systematic pattern of switching between the two target positions (i.e., differences in classification performance for left-right versus top-bottom). Second, also a trial-based correlation analysis of classifier confidence scores showed the absence of a negative correlation between the target positions. Our findings are therefore most consistent with a limited-capacity parallel model, in which observers maintain two templates active during search, but with mutually aversive consequences. However, we point out that our data do not exclude the possibility of seriality. First, while there may have been little seriality in selecting the targets from the displays on the basis of color, there may have been a serial component in accessing their alphanumeric identity — a component to which our classifier was not sensitive. Moreover, there is still a distinct possibility that imposing seriality is a valid strategy

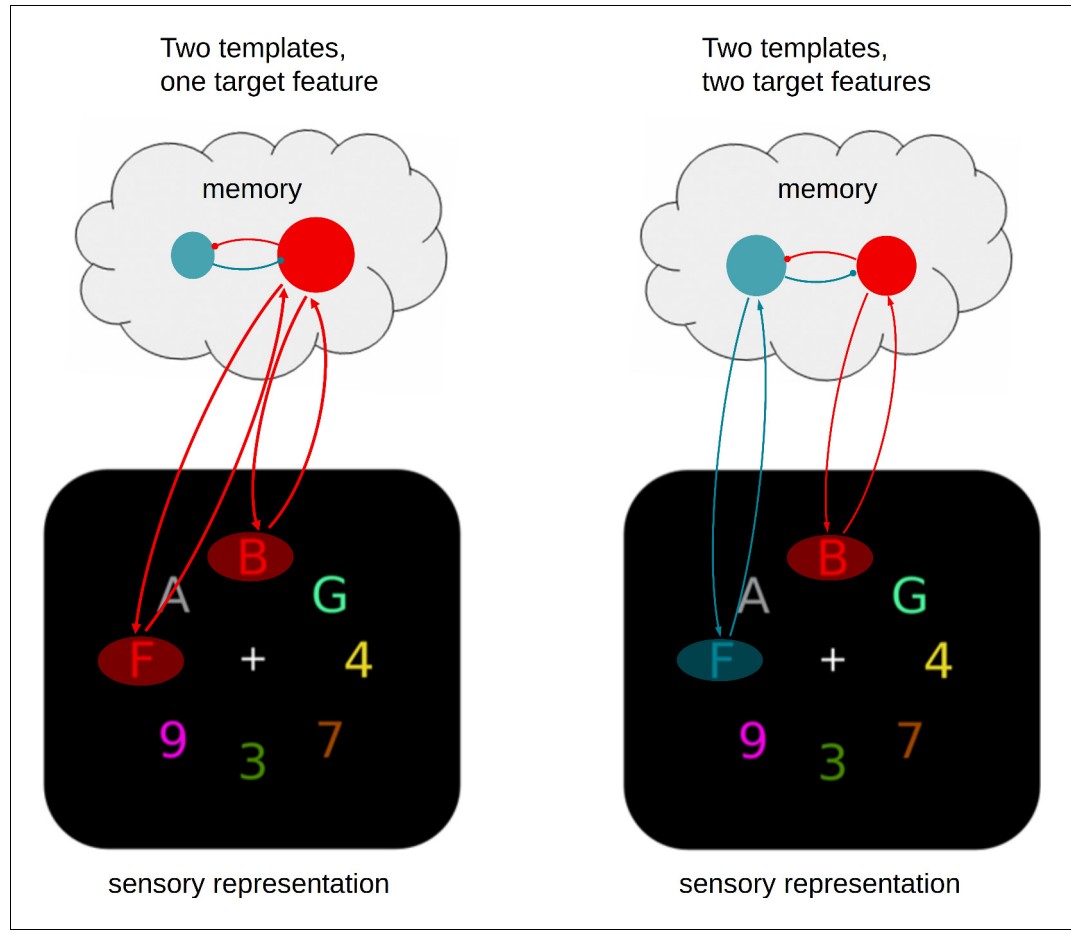

**Figure 4.** A limited parallel model. Attentional templates in memory engage in recurrent feedback loops with matching sensory representations, resulting in target enhancement. Multiple templates can be activated in parallel and may be equally active prior to search, but they compete through mutual suppression, which has consequences during search. (A) The presence of a single target feature in the sensory input will unequivocally trigger one of the active templates, eventually resulting in as strong selection as when there is only one template (not shown), albeit at a short delay. (B) When both templates are activated the mutual suppression will prevent strong activation of either, resulting in substantially weakened and delayed selection of both targets.
DOI: https://doi.org/10.7554/eLife.49130.016

that observers may deploy to resolve competition between different target features, but that such choices depend on tasks, context, or instructions (*Cave et al., 2018*; *Stroud et al., 2019*). For example, we previously observed evidence for serial switching in a different paradigm when observers had to select only one of two targets present, and were instructed to switch at least a few times during a block (*Ort et al., 2019*; *van Driel et al., 2019*). The current results indicate that the process can occur in parallel, not that it must.

Our account has a resemblance to the Boolean Map Theory of Visual Attention (*Huang and Pashler, 2007*), which proposes a division of attentional selection into two components: (1) The feature-to-location routine, in which task-relevant features are being located in the visual field (referred to by Huang and Pashler as *selection,* but here analogous to what we call preparation for selection) and (2) the location-to-feature routine in which individuals extract response-relevant features at a target location (*access,* analogous to what we here call engagement in selection). However, in contrast to the model that we propose, in which only what Huang and Pashler refer to as the access aspect of search is severely limited, the Boolean Map theory poses a capacity limitation of one on both selection *and* access.

We believe the distinction between template preparation and template engagement in selection has great potential for resolving the current debate on whether observers can look for more than a single target at the same time (*Menneer et al., 2009*; *Beck et al., 2012*; *Irons et al., 2012*; *Grubert and Eimer, 2015*; *Beck and Hollingworth, 2017*; *Ort et al., 2017*; *Ort et al., 2018*). Studies central to this debate have largely focused on how many templates can be prepared in anticipation for a search, rather than how many of these templates can then be concurrently engaged in selection without costs. From our data, the answer to the question then appears to be yes, observers may *look* for multiple targets simultaneously at little cost, but it is *selecting* those targets that runs into real limitations.

Although we found the costs of going from one to two templates to be relatively small, this leaves open the question whether costs will increase more strongly with more templates being added. As there is more opportunity for memory representations to interfere with each other when multiple memory representations need to be maintained, this would be expected. Such interference will depend on the similarity of the to-be-remembered templates, as well as the assumed capacity. Although the capacity of VWM is thought to be around three to four items (at least for the standard colored shapes used in experiments like ours), and VWM is assumed to be central to top-down driven search, there is ample evidence that visual search needs not solely rely on VWM. In fact, given that in our experiment the target template remained the same for a block of trials, observers may have at least partly relied on trained templates in long term memory here (*Carlisle et al., 2011*; *Gunseli et al., 2014*). Moreover, work by *Wolfe (2012)* has shown that observers can successfully search for any one of tens of different target objects if given the opportunity to first commit these objects to long term memory. In our study, effects of capacity limits and any interference arising from it may be have been stronger when the targets would have been cued from trial to trial, rather than from block to block. Conversely, the fact that we find limitations even with repeated targets is testament to the mechanistic bottleneck in the engagement of selection that we propose. One reason may be that even LTM representations would need to become activated for effective task-or context-driven guidance (since only one set of trained colors is relevant in a particular block). Our data suggests that the limit may well be in this goal-driven aspect of search – that is, the deployment of a representation for perceptual bias rather than its storage per se. In line with this, *Grubert et al. (2016)* reported evidence that attentional selection per se, as measured by the N2pc (which at least partly underlies the signal also used here), is not affected by whether targets are stored in long term memory or working memory. Future research will need to shed further light which memory systems support search templates.

Finally, the question of memory capacity or interference is also important when considering that current limitations were found when both target features were drawn from the same dimension (color). There is evidence that different dimensions may to some extent independently store (e.g., *Wang et al., 2017*), or guide attention towards (*Wolfe, 1994*; *Jenkins et al., 2017*), target features. Our methods may therefore prove useful in assessing the exact limitations of selecting targets defined along different dimensions.

To sum up, we propose that models of visual selection need to consider the difference between preparing for selection and engaging in selection of multiple visual targets. We demonstrate that whereas the first process comes at little cost, the true bottleneck of multiple-target selection is in engaging multiple template representations.

## Materials and methods

### Materials availability

All data and material will be made freely accessible at https://osf.io/3bn64.

### Participants

Thirty-two participants naive to the purpose of the experiment were recruited at the Vrije Universiteit Amsterdam and were compensated with money or course credit. Eight were excluded due to poor behavioural performance in at least one experimental condition (a predefined cutoff of accuracy <85% was used, see below) to ensure sufficient numbers of correct trials for the EEG analyses. The remaining twenty-four participants (age: 19–30 years, M = 22.0; 17 females, seven males) had

normal or corrected-to-normal visual acuity and color vision. All participants gave written informed consent in line with the Declaration of Helsinki. The study was approved by the Scientific and Ethics Review Board of the Faculty of Behavioural and Movement Sciences at the Vrije Universiteit Amsterdam (Reference number: VCWE-2016–215).

## Stimuli and procedure

Displays consisted of eight colored alphanumerical characters evenly spaced on an imaginary annulus with a radius of 2.5 degree visual angle (dva), centered at the middle of the screen (*Figure 1A*). The characters were uppercase letters (K, H, M and T) and digits (7, 6, 3 and 4, each spanning approximately 1.2 dva vertically and between 0.8 and 1.0 dva horizontally. In total, eight colors were used in the experiment: Red (RGB-values: 224, 0, 38), green (0, 155, 0), blue (55, 110, 255), and yellow (160, 95, 5) were potential target colors (all approximately isoluminant, $\sim$21 cd/m$^2$, min-max range: 19–25 cd/m$^2$), whereas purple (145, 30, 180), cyan (70, 240, 240), pink (250, 0, 179) and gray (130, 130, 130) were always used as distractor colors (M = 35 cd/m$^2$, min-max range: 16–63 cd/m$^2$). The stimuli were presented on a black background (0, 0, 0).

Participants were instructed to find two color-defined target characters on each trial, and indicate whether or not these belonged to the same alphanumerical category (i.e., letters or numbers). Response keys were counterbalanced across participants. In the beginning of a block, the task-relevant colors were shown to the participants as two target-colored disks (spanning 1.2 dva each), for 2000 ms. Depending on the experimental condition (see below) either one colored disk was presented in the middle of the screen, or 1.0 dva to the left and right of the center, respectively. The target colors were valid for a block of 64 trials after which new colors were shown. Throughout the trial a white fixation cross remained visible in the middle of the screen which participants were required to keep fixating. The trial sequence began with a fixation screen presented for 850 to 950 ms (randomly selected from a uniform distribution), followed by a search display for 50 ms and another fixation screen until a response was given or a 2000 ms timeout. Finally, a written message ('correct' or 'wrong') presented for 500 ms indicated whether the response was correct or not. In case participants did not respond before the timeout, the experiment was paused for ten seconds to encourage them to respond quicker henceforth. After every block, participants received feedback on accuracy. Note that for the first two participants presentation time was two display frames (~16.67 ms) shorter than for the rest of the sample. To facilitate good behavioral performance, we increased presentation time from the third participant onwards. However, as these two participants performed well, even with 16.7 ms shorter presentation rates (and thus met our inclusion criteria), we decided to keep them in the sample.

One target color was always presented on the horizontal axis (left or right of fixation), while the other was always presented on the vertical axis (above or below fixation), with color-position assignment randomly chosen but occurring equally often. Participants were informed that targets would appear only on the cardinal axes of the search array. The irrelevant items on the diagonals were added to the search display to increase competition, increase color heterogeneity, and to prevent participants from looking for any color duplicates rather than for the specific target color, whenever both target objects of a search array shared the same color (as was the case in the one target feature conditions). To further prevent participants from employing the strategy of selecting color duplicates or groups, rather than setting up a template for the specific target color, one half of all trials, one of the additional distractor colors was duplicated and presented at one of the diagonal positions. In doing so, the mere presence of a duplicated color would not signal these to be the target items, so that a color-specific would be necessary to perform the task efficiently. One target color was always presented on the horizontal axis (left or right of fixation), while the other was always presented on the vertical axis (above or below fixation), with color-position assignment randomly chosen but occurring equally often. The alphanumerical identity of each search item was chosen randomly with the restriction that the two target objects belonged as often to the same category (both letters or both digits) as to different categories (one letter and one digit). Consequently, alphanumerical category and positions of both targets were fully counterbalanced within a block.

## Design

Across blocks, we introduced three experimental conditions that differed in (1) how many colors were task-relevant (i.e., the number of templates, TMP) and (2) how many target colors appeared in a single search display (i.e., the number of different target features, TGT). In *1TMP–1TGT* blocks, only one color was task-relevant, so that both target characters had the same color and participants knew beforehand which color they would need to select. In the *2TMP–1TGT* block type, two colors were cued as task-relevant, but only one of the two target colors would actually appear in a search display, as was randomly determined from trial to trial (with equal numbers for each target color). Participants could not predict which of the two target colors would be present in a specific search display, therefore they had to keep two templates active, even though only a single color was required for selecting the actual targets. Finally, in the *2TMP–2TGT* block type, again two colors were cued as task-relevant, but now both these target colors also appeared in each search display, so that both colors were required for selection. Each condition was repeated eight consecutive times.

We decided to use a blocked design rather than intermixing trials of all three conditions within blocks because pilot data indicated that behavioral performance is rather low if conditions are mixed within a block, so that many trials would have to be excluded. Furthermore, intermixing conditions would also make it necessary to cue not only the condition, but also the task-relevant colors before every trial. This would have increased the duration of a trial, and hence reduced the total number of trials that we could fit in a session, thus reducing power even more.

When only one color was task-relevant (1TMP–1TGT), each of the four colors would thus serve as the target color twice, whereas in blocks in which two colors were task-relevant (2TMP–1TGT and 2TMP–2TGT), observers would look for the combinations red and green or blue and yellow, each four times. We chose these color combinations as they are not linearly separable in color space and thus prevented participants from potentially setting up a single template encompassing both target features. The order of conditions was counterbalanced across participants. Prior to the start of the experiment, participants received instructions and practiced all conditions in increasing order of difficulty (1TMP–1TGT, 2TMP–1TGT, 2TMP–2TGT). During practice, participants repeated blocks of 32 trials for each condition as often as necessary to reach an accuracy of 85%, but at least three times. Note, even if participants had initially reached this inclusion criterion, they might still have performed below 85% during the experiment. Therefore, eight participants with an accuracy below 85% were excluded from the analysis.

## Apparatus and EEG acquisition

The experiment was designed and run using the OpenSesame software package (version 3.2.2; *Mathôt et al., 2012*). Stimuli were presented on a 22-inch Samsung Syncmaster 2233 monitor, with a resolution of 1680 × 1050 pixels at a refresh rate of 120 Hz. Participants were seated in a dimly lit, sound-attenuated room in a distance from the screen of approximately 70 cm and the eyes aligned with the center of the screen. A QWERTY PS/2 keyboard was placed in the lap of each participant. They were instructed to place left and right index fingers on the *z* and *m* keys to indicate whether targets were of the same or different category. Further, they were asked to refrain from excessive blinking and motion during the experiment. The experimenter received real-time feedback on behavioral performance and quality of EEG recording in an adjacent room.

We used the BioSemi ActiveTwo system (Biosemi, Amsterdam, The Netherlands) to record from 64 AG/AgCl EEG channels, four EOG channels and two reference channels at a sampling rate of 512 Hz. EEG channels were placed according to the 10–20 system. EOG channels were placed one cm outside the external canthi of each eye to measure horizontal eye movements and two cm above and below the right eye, respectively to measure vertical eye movements and blinks. Reference electrodes were placed on the left and right mastoids.

## EEG preprocessing

All EEG preprocessing and analyses were performed offline in Matlab (2014b, The Mathworks) and Python (2.7, www.python.org), using a combination of EEGLAB (*Delorme and Makeig, 2004*), the Amsterdam Decoding And Modeling toolbox (ADAM, version: 1.07-beta, *Fahrenfort et al., 2018*) and custom scripts (freely accessible at https://osf.io/3bn64). EEG data were first re-referenced to

the average of the left and right mastoids. No offline filters were applied to the data. Next, the continuous signal was split into epochs from 300 ms before until 800 ms after search display onset. Epochs were baseline corrected by removing the average activity in a pre-stimulus window between −100 and 0 ms from each time point. All epochs in which participants failed to respond correctly, or response times were lower than 200 ms (anticipatory errors) or greater than three standard deviations above the block mean were removed from further analyses (mean exclusion: 6.6%, min-max range across participants: 4.4–9.7%). To make sure that the EEG would not be contaminated by eye movements, we scanned epochs for horizontal eye movements within the first 500 ms after stimulus onset (amplitude threshold: 30 μV, window length: 100 ms, step size: 50 ms) and removed epochs containing such. This resulted in an exclusion of on average 2.4% (min-max range: 0.0–16.8%) of all epochs. Noise due to muscle activity was removed using an automatic trial-rejection procedure. To specifically capture EMG, we used a 110–140 Hz band-pass filter, and allowed for variable z-score cut-offs per participant based on the within-subject variance of z-scores, resulting in the exclusion of on average 7.1% (min-max range: 1.8–13.1%). Next, all epochs were visually inspected for any obviously contaminated trials that have been missed by the automatic trial-rejection procedure (mean exclusion: 0.4%, min-max range: 0.0–0.9%). To identify and remove components related to blinks, we used EEGLAB's implementation of independent component analysis (ICA). In total, 15.6% (min-max range: 8.1–34.2%) of all epochs were removed during preprocessing.

## Decoding of target locations

The main analyses decoded target positions based on the raw EEG of all 64 channels, using the ADAM toolbox (*Fahrenfort et al., 2018*). To that end, we used a 10-fold cross-validation scheme by splitting the data of individual participants into ten equal-sized folds after randomizing the order in which trials occurred in the experiment. A linear discriminant classifier was then trained on the data of nine folds and tested on the data of the tenth one. This procedure was repeated ten times until each fold served as a test set once. Finally, the classifier performance was averaged across all individual folds. For each condition, we trained one classifier to differentiate trials on which one of the targets was presented on the left versus the right position, and another classifier to differentiate the same trials as to whether the other target was presented on the top versus the bottom position. Furthermore, to account for minor incidental imbalances with respect to the trial count per class introduced by the trial rejection procedure, we performed within-class and between-class balancing. For within-class balancing, we undersampled trials to match the number of trials in which the target appeared on the irrelevant dimension within each class. For example, when training a classifier to differentiate trials in which the target appeared on the left versus right position, we made sure that each class (e.g., left targets) contained the same number of trials in which the second target appeared on the top or the bottom position by removing trials of the more frequent trial type. Between-class balancing entailed the oversampling of trials (generating synthetic samples based on the existing data; see *He, 2008*) belonging to the less frequent class, so that the classifier would not become biased toward the more frequent class. As performance measure we used the Area Under the Curve (AUC, *Hand and Till, 2001*), which is an unbiased measure that is based in signal detection theory and describes the area under the receiver-operator curve when plotting hit rate over false alarm rate. The decoding performance for single conditions was statistically tested against chance level (AUC = 0.5) by running two-sided one-sample $t$-tests across participants for every time point, or by testing AUCs against each other when comparing conditions directly. To correct for multiple comparisons, we used cluster-based permutation tests (5000 permutations) on adjacent time points with the alpha level set to $\alpha = 0.05$ (*Maris and Oostenveld, 2007*). Next, to examine the topography of the activations, we multiplied the classifier weights across all channels with the data covariance matrix, yielding activation maps that can directly be interpreted as neural sources (*Haufe et al., 2014*).

## Estimating and statistically testing onset latency and amplitude difference

To estimate the onset latencies at which target location became decodable, we used an approach in which we combined computing the fractional peak latency with a jackknife-based approach (*Miller et al., 1998*; *Luck, 2014*; *Liesefeld, 2018*). Group-averaged classification scores were

repeatedly computed over all but one participant, until each participant was left out once. For each of these averages, we estimated its onset latency by identifying the peak amplitude in the window of 150 to 700 ms after search display onset and defined the onset as the first point in that time window in which the classification scores exceeded 50% of the peak score. To mitigate the influence of high-frequency noise on the latency estimation, for every time point we averaged the amplitude of that time point and the two adjacent time points for peak and onset latency estimation. To statistically test for potential differences in onset latencies across experimental conditions, we followed *Miller et al. (1998)* and computed *t*-statistics for the pairwise comparisons between the 1TMP–1TGT and 2TMP–1TGT condition, and between the 2TMP–1TGT and 2TMP–2TGT condition. The procedure corrects for the artificially reduced error term due to the jackknifing by effectively dividing the *t*-statistic by the degrees of freedom.

Finally, the 2TMP–1TGT condition (when two target colors were cued but only one of them was present in any one search display) allowed us to asses intertrial switch costs (*Maljkovic and Nakayama, 1994*; *Olivers and Humphreys, 2003*; *Wolfe et al., 2004*; *Dombrowe et al., 2011*; *Ort et al., 2017*), by splitting the 2TMP–1TGT condition into repeat and switch trials and to run all analyses separately for these two trial types.

## N2pc analysis

Even though the backward decoding approach would already show whether and when location-specific information would be present in the raw EEG, for the sake of comparison to the existing N2pc literature, we also conducted a more common event-related potential (ERP) analysis to examine latency and amplitude of the N2pc component. First, to identify N2pc components, we computed ERPs locked to stimulus onset at electrodes PO7 and PO8. ERPs at the ipsilateral electrode relative to the horizontal target position (i.e., PO7 for targets on the left, PO8 for targets on the right), were subtracted from ERPs at the contralateral electrode, collapsed over the vertical target position, but separately for each participant and condition. The resulting difference wave forms were then statistically tested against zero with two-sided one-sample *t*-tests at each time point. A cluster-based permutation test (5000 permutations, $\alpha = 0.05$) was performed on contiguous time points to correct for multiple comparisons (*Maris and Oostenveld, 2007*). To quantify amplitudes and onset latency the same approach as for the classification scores was used, with the exception that we did not use the entire epoch when looking for the peak, but only the window of 200–350 ms post stimulus, as this is the time window in which the N2pc is typically observed (e.g., *Eimer, 1996*; *Eimer and Grubert, 2014*).

## Correlating classification confidence

Another useful feature of the AUC measure is that it considers the confidence that a classifier has about class membership of a particular instance at every time point. Confidence is expressed as the distance from the decision boundary and can be interpreted as the representativeness of a certain instance (EEG activity across all channels for a given time point) of that class (*Ritchie and Carlson, 2016*; *Grootswagers et al., 2018*). Applied to the present paradigm, we assumed that the more strongly prioritized a particular target position, the higher the classifier's confidence scores. Based on this logic, we reasoned that if prioritization is limited to a single target at a time, a classifier cannot simultaneously have high confidence about both targets, and thus confidence should correlate negatively. To test this hypothesis, we extracted the confidence scores of both classifiers (left-right and top-bottom) and correlated these across trials (*Spearman's ρ*), separately for each time point and condition. If prioritization is limited to a single item for certain time points, we would expect a moderately negative correlation between left-right and top-bottom classifiers for those points, because whenever the classifier has high confidence in one dimension, it will have low or random confidence in the other dimension, and vice versa. If on the other hand, prioritization occurs in parallel and selection strength is driven by a common mechanism, one would expect a positive correlation at those time points. Finally, if prioritization occurs in parallel but selection strength is driven by independent mechanisms, one would expect zero correlation. To assess these competing hypotheses, correlations were statistically tested against zero by running two-sided one-sample *t*-tests across participants at every time point, using cluster-based permutation tests (5000 permutations, $\alpha = 0.05$) to correct for multiple comparisons (*Maris and Oostenveld, 2007*).

## Correlating classification confidence on simulated data with known underlying correlational structure

When correlating confidence scores, a lack of correlation could reflect parallel processing of the two targets, but could also be caused by the decoding strength being too weak, due to an insufficient signal-to-noise ratio (SNR) in the data. To make sure that we had enough statistical power to detect a correlation if it was actually present, we ran a simulation in which we embedded a signal in systematically manipulated noise levels, and determined at which decoding strength a known correlation could be reliably extracted. The same analysis pipeline was applied as for the actual data. Specifically, we replaced the data of eight channels with simulated data in which we injected either a positive, negative or null correlation between horizontal and vertical targets and varied the overall noise level. The data were created by generating half a cycle of a sine wave with an amplitude of 1 μV, extending over 400 ms (200–600 ms post stimulus) and assigned to a subset of channels to reflect attentional selection. To create location-specific effects (i.e., contra vs. ipsilateral), we injected the same ERP with a negative amplitude on an orthogonal subset of the channels. Therefore, attentional selection was simulated with a positive ERP on half the channels and a negative ERP on the other half. Importantly, attentional selection of vertical and horizontal targets was simulated independently, by using an orthogonal split of the channels into contra- and ipsilateral. For every correlation pattern we simulated 512 trials, the same number as in the real experiment. For the positive correlation, we injected the ERP for both vertical and horizontal targets on half of the simulated trials, and no ERP on the other half, reflecting either both targets to be selected simultaneously, or none of them (i.e. parallel selection). For the negative correlation, the ERP was either injected for vertical targets or for horizontal targets (each half of the trials), but never in both, reflecting the selection of either one or the other target (i.e. serial selection). For the null correlation, per trial, we randomly chose whether an ERP was present for one of the targets, both, or none. Next, we added random noise for all trials. Critically, the SNR was parametrically manipulated, relative to the (constant) amplitude of the ERP. For example, a SNR of 4 means the peak ERP amplitude was four times as high as the maximum noise amplitude. In total, we used SNRs of 4, 2, 1.33, 1, 0.67, 0.5, 0.33, 0.25, 0.2, 0.17, 0.14, 0.13, 0.11, 0.1, 0.07, 0.05, and 0.04. Once the simulated dataset was created, the same backward decoding model (see Materials and methods) was used to decode the target location, separately for the vertical and horizontal target, the injected correlation and the SNR. Similarly, the classifiers' confidence scores were correlated between vertical and horizontal targets, as was done for the actual data.

## Additional information

### Funding

| Funder | Grant reference number | Author |
|---|---|---|
| Nederlandse Organisatie voor Wetenschappelijk Onderzoek | 464-13-003 | Christian N L Olivers |
| H2020 European Research Council | ERC-2013-CoG-615423 | Christian N L Olivers |

The funders had no role in study design, data collection and interpretation, or the decision to submit the work for publication.

### Author contributions

Eduard Ort, Conceptualization, Data curation, Software, Formal analysis, Investigation, Visualization, Methodology, Writing—original draft, Project administration, Writing—review and editing; Johannes Jacobus Fahrenfort, Conceptualization, Software, Formal analysis, Supervision, Methodology, Project administration, Writing—review and editing; Tuomas ten Cate, Formal analysis, Investigation, Methodology, Writing—review and editing; Martin Eimer, Conceptualization, Writing—review and editing; Christian NL Olivers, Conceptualization, Supervision, Funding acquisition, Methodology, Writing—original draft, Project administration, Writing—review and editing

## Author ORCIDs

Eduard Ort (iD) https://orcid.org/0000-0001-5546-3561
Johannes Jacobus Fahrenfort (iD) https://orcid.org/0000-0002-9025-3436
Christian NL Olivers (iD) https://orcid.org/0000-0001-7470-5378

## Ethics

Human subjects: All participants gave written informed consent in line with the Declaration of Helsinki. The study was approved by the Scientific and Ethics Review Board of the Faculty of Behavioural and Movement Sciences at the Vrije Universiteit Amsterdam (Reference number: VCWE-2016-215).

## Decision letter and Author response

Decision letter https://doi.org/10.7554/eLife.49130.021
Author response https://doi.org/10.7554/eLife.49130.022

# Additional files

## Supplementary files

• Transparent reporting form
DOI: https://doi.org/10.7554/eLife.49130.017

## Data availability

All data and material will be made freely accessible at https://osf.io/3bn64.

The following dataset was generated:

| Author(s) | Year | Dataset title | Dataset URL | Database and Identifier |
|---|---|---|---|---|
| Ort E, Fahrenfort JJ, ten Cate T, Eimer M, Olivers CNL | 2019 | Data from: Humans can efficiently look for but not select multiple visual objects | https://osf.io/3bn64 | Open Science Framework, 3bn64 |

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
