## [Decision Letter]

Thank you for submitting your article "Humans can efficiently look for, but not select multiple visual objects" for consideration by *eLife*. Your article has been reviewed by three peer reviewers, one of whom is a member of our Board of Reviewing Editors, and the evaluation has been overseen by a Reviewing Editor and Joshua Gold as the Senior Editor. The following individuals involved in review of your submission have agreed to reveal their identity: Liqiang Huang (Reviewer #2); Hui Chen (Reviewer #3).

The reviewers have discussed the reviews with one another and the Reviewing Editor has drafted this decision to help you prepare a revised submission.

Summary:

This work used an interesting paradigm in combination with EEG recordings and multivariate decoding approach to dissociate the multi-template preparation and selection process. They found only small neural and behavioral costs associated with simultaneously maintaining two templates but substantial costs when engaging the templates in selection. The work addresses a hotly debated question about the limitation of selecting multiple targets and provides novel evidence and viewpoints. There are still some major issues brought up by the reviewers that need the authors to address and do additional analysis.

Essential revisions:

1) About the block design. Instead of being randomly mixed, the three conditions were presented in separate blocks. This would introduce long-term memory effects and different attentional or search strategy for each condition. According to previous research, when the search target remained the same per trial within a block, long-term memory representations but not working memory representations came to drive the attentional selection (Carlisleet al., 2011; Gunseli, Olivers and Meeter, 2014; Gunseli, Olivers and Meeter, 2016; Dirk, Jan and Olivers, 2016). The authors should add more clarifications and discussions about the limitation of the design and its associated interpretations (e.g., involvement of long-term memory).

2) The main results are less (but still significant?) cost for 2TMP-1TGT compared to 1TMP-1TGT (preparation) and larger cost for 2TMP-2TGT compared to 2TMP-1TGT (selection). However, all the statistical results are based on separate comparisons and there is no direct comparison between the two type of task effects. To support the claim that the preparation cost is less than the selection cost, direct statistical analysis comparing the two effect is needed.

3) About the task. The participants were asked to not only select the template-matching target but also judge whether they came from the same category. Therefore, the cost might be caused only (or partly) by the comparison of two targets from two different colors, instead of deriving from the selection stage. In other words, the cost might still emerge in a comparison task that does not include the search/selection process and thus has nothing to do with the template selection. Actually, previous work (e.g., Boolean map theory) has shown that it is more difficult to access two different colors in comparison with two same color. The authors should address the issues in the revision.

4) In the subsection “Stimuli and Procedure”, the authors mentioned that on half of the trials they added one more color duplicate on the diagonals to prevent participants from just looking for any color duplicates. The description of the location of the added color duplicates is ambiguous. Do they appear on the four locations aside from vertical and horizontal axes? If so, this manipulation would not work because participants only need to pay attention to four locations on the vertical and horizontal axes.

5) The final analysis to test whether the selection cost is due to competition or serial selection is interesting, but the reviewers are not fully convinced by the null results. The analysis seems to be based on a quite strong hypothesis that the two targets would show consistent negative correlation across time and trials. Meanwhile, it is quite possible that their competition might only appear at some time points within a trial rather in a continuously alternative manner. The authors should discuss relevant limitations of the analysis in the revision.

---

## [Author Response]

Summary:This work used an interesting paradigm in combination with EEG recordings and multivariate decoding approach to dissociate the multi-template preparation and selection process. They found only small neural and behavioral costs associated with simultaneously maintaining two templates but substantial costs when engaging the templates in selection. The work addresses a hotly debated question about the limitation of selecting multiple targets and provides novel evidence and viewpoints. There are still some major issues brought up by the reviewers that need the authors to address and do additional analysis.Essential revisions:1) About the block design. Instead of being randomly mixed, the three conditions were presented in separate blocks. This would introduce long-term memory effects and different attentional or search strategy for each condition. According to previous research, when the search target remained the same per trial within a block, long-term memory representations but not working memory representations came to drive the attentional selection (Carlisleet al., 2011; Gunseli, Olivers and Meeter, 2014; Gunseli, Olivers and Meeter, 2016; Dirk, Jan and Olivers, 2016). The authors should add more clarifications and discussions about the limitation of the design and its associated interpretations (e.g., involvement of long-term memory).

This is indeed an issue that deserves more attention than we originally devoted to it. Specifically, the reviewers raise the possibility (A) that the repeated presentation of the same template features led to attentional guidance being controlled by long-term memory (LTM), rather than working memory (WM) representations, and (B) that participants employed different search strategies in each of the conditions as they knew what condition to expect.

As for (A), it is true that we do not actually know whether it is WM or LTM that drove attentional selection. For this reason, we chose to not make unsubstantiated claims about which memory system is (solely) involved in our paradigm and remove all reference in the manuscript to WM-driven guidance as the single mechanism. However, we would argue that guidance would at least be based on activated LTM, as it concerns a goal-driven process, and as such would be in line with WM representing the activated part of LTM. We now dedicated more room in the discussion to cover the issue of WM vs. LTM-driven attentional guidance (Discussion section). Nevertheless, the effects we describe and the conclusions we draw, are largely distinct from the issue of the specific memory system involved. We meant to test which aspect of multiple-target search is limited, particularly, whether preparation for or engaging in selection, if anything, is the bottleneck. Accordingly, the most relevant time window, in which we also find most of the effects, is the N2pc time window (200-300 ms post stimulus). Importantly, there is also considerable evidence, that whether attentional guidance is based on WM or LTM does not change the latency and magnitude of the N2pc (Grubert, Carlisle and Eimer, 2016; Gunseli, Olivers and Meeter, 2014). Beyond that, what the observed limitations tell us about WM and LTM was only of secondary interest.

As for (B), when designing the study, we considered intermixing trials of all three conditions within blocks, and then cueing the specific condition on each trial. However, we decided against it because such a procedure demands a lot from subjects in terms of task compliance and pilot data indicated that behavioral performance is rather low in that case, so that many trials would have to be excluded. Furthermore, intermixing conditions would also make it necessary to cue not only the condition, but also the task-relevant colors before every trial. This would have increased the duration of a trial, and hence reduced the total number of trials that we could fit in a session, thus reducing power even more. We now also explain our motivation for using a blocked-design in subsection “Stimuli and Procedure” in the manuscript.

We cannot exclude the possibility that participants used different strategies, but if so, we would assume that they would choose the strategy that would optimize performance – in which case the limitations in selection that we reveal become even more informative. Note that the brief presentation of the displays made it necessary to quickly extract the important information in all conditions. The only variability here could have arisen by participants selecting based on perception vs. from memory. The data suggest that participants were not relying solely on memory (otherwise no N2pc would have been observed), though subjects may have had to rely more on memory when selection was slow (in the 2 templates 2 targets case, which again would be meaningful). Therefore, we do not think that different search strategies as such challenge the validity of our conclusions.

2) The main results are less (but still significant?) cost for 2TMP-1TGT compared to 1TMP-1TGT (preparation) and larger cost for 2TMP-2TGT compared to 2TMP-1TGT (selection). However, all the statistical results are based on separate comparisons and there is no direct comparison between the two type of task effects. To support the claim that the preparation cost is less than the selection cost, direct statistical analysis comparing the two effect is needed.

In the manuscript, we claimed that selection cost is greater than preparation cost based on the behavioral findings and EEG findings. Even though we directly compared these two sources of multiple-target search cost for the behavioral data (see subsection “Behavioral results”), the reviewers correctly point out that a similar analysis of the EEG data was missing. Therefore, to back up our claim with a direct statistical analysis, we subtracted the time course reflecting preparation costs from the time course reflecting selection costs (i.e. [2TMP-1TGT – 2TMP-2TGT] – [1TMP-1TGT – 2TMP-1TGT]) and ran a cluster-based permutation on the resulting time course. This analysis indeed revealed two temporal clusters in which the selection cost was greater than the preparation cost, whereas no cluster showing the inverse pattern was observed. Based on this finding, we are confident that our claim is supported by the data. We added this analysis to the manuscript (see subsection “Decoding of target positions based on the raw EEG”).

Please note further that while re-running the analysis to directly test these conditions, we noticed that we had provided incorrect numerical values for the 2TMP-1TGT condition. We corrected these. Importantly, all conclusions still hold, as the results did not change qualitatively.

3) About the task. The participants were asked to not only select the template-matching target but also judge whether they came from the same category. Therefore, the cost might be caused only (or partly) by the comparison of two targets from two different colors, instead of deriving from the selection stage. In other words, the cost might still emerge in a comparison task that does not include the search/selection process and thus has nothing to do with the template selection. Actually, previous work (e.g., Boolean map theory) has shown that it is more difficult to access two different colors in comparison with two same color. The authors should address the issues in the revision.

It is true that participants needed to not only find and select both task-relevant objects, but also compare their alphanumeric category. However, we do not believe that this aspect of the task contributed to the observed effects, for two reasons. First, the comparison of the categories had to be done in all three conditions. Therefore, if the comparison process would invoke any cost, it should be manifested in all three conditions. Second, the selection cost was observed already early in a trial (starting around 200 ms post stimulus, see Figure 2). This time window has been shown to be sensitive to color-based selection, whereas semantic information about alphanumerical category usually emerges later (e.g. Kutas and Federmeier, 2000). Therefore, it is unlikely that the observed selection cost is caused by comparisons processes relying on semantic information rather than the template selection process.

Finally, we agree that our data and proposed model is quite similar to the Boolean Map Theory (Huang and Pashler, 2007). Particularly, the rather severe capacity limitation of access to objects in the visual field is striking. Nevertheless, there are also some differences. For example, according to the Boolean Map theory, also the preparation for selection aspect of search is limited to a single feature within a feature dimension (i.e. a single color), whereas our data suggest that this aspect is associated with only a little cost. We therefore decided to discuss Boolean Map theory more extensively in the manuscript (Discussion section).

4) In the subsection “Stimuli and Procedure”, the authors mentioned that on half of the trials they added one more color duplicate on the diagonals to prevent participants from just looking for any color duplicates. The description of the location of the added color duplicates is ambiguous. Do they appear on the four locations aside from vertical and horizontal axes? If so, this manipulation would not work because participants only need to pay attention to four locations on the vertical and horizontal axes.

According to the reviewers’ suggestion we improved the clarity of that section in the manuscript (see subsection “Stimuli and Procedure”).

Indeed, the additional duplicate colors could only appear in on the diagonal positions, that is, not on any of the potential target positions. We agree with the reviewer that participants might in principle focus on just the target positions. However, we can tell from experience that attending to four potential target locations simultaneously under 50 ms duration, of which then two need to be ignored while trying to detect a duplicate at the other locations proves to be a rather taxing task, and that simply attending to the target feature value is a much more intuitive approach. The addition of duplicate distractor colors mainly served to further prevent grouping as such as a cue for selection (in addition to interleaving the targets with distractors on the diagonals). Consistent with this view, fMRI evidence has shown that even when participants actively engage only with a single axis during visual search (e.g. horizontal), they still activate representations matching their template on the other axis (e.g. vertical) even when this axis is fully task irrelevant. This shows that the brain does not seem to be able to effectively suppress part of the visual field during search (Peelen, Fei-Fei and Kastner, 2009).

5) The final analysis to test whether the selection cost is due to competition or serial selection is interesting, but the reviewers are not fully convinced by the null results. The analysis seems to be based on a quite strong hypothesis that the two targets would show consistent negative correlation across time and trials. Meanwhile, it is quite possible that their competition might only appear at some time points within a trial rather in a continuously alternative manner. The authors should discuss relevant limitations of the analysis in the revision.

First of all, to preclude any misunderstandings, we would like to clarify that the analysis presented in Figure 3 would, in principle, have been able to show effects of competition also for only a few consecutive time points. For every time point, we correlated the confidence of AUC scores across trials. Therefore, if there were time points that would have shown a systematic relationship between horizontal and vertical targets, we should be able to identify these time points, even if only for a single sample. In fact, as can be seen in Figure 3, a short time period around 500 ms post stimulus showed a significant positive correlation between horizontal and vertical targets for the 2TMP-2TGT condition (even though we believe this effect to be spurious, given its negligible size).

That said, we are the first to admit that the presented analysis is at the limits of what can be done with the data. It is true that if the relationship is not consistent across trials and time, the analysis might have not been sensitive enough to pick up this relationship. For example, if there was an actual negative correlation over a very short time window (as would be predicted by a serial model), but its precise timing varied across the trials, the average correlation would probably be strongly reduced. There may be even very rapid switching back and forth. However, we would argue that any such very brief serial moments (or ultra fast switching) would always be very hard to distinguish from a parallel process, and may be theoretically less relevant in the context of the debate, which revolves around the argument of relatively slow (i.e. measurable) serial mechanisms. Nevertheless, according to the reviewers’ suggestion, in the manuscript (see subsection “Sample-wise correlation of classifier confidence across trials as a measure of inter target dependency”) we more explicitly point out the potential limitations of our analysis.

Again, we wish to be forthright here, and in the manuscript that this effect is essentially a null effect, and thus difficult to interpret. We wish to point out that we did not observe any evidence in favor of a serial account (the weak correlations were not even in the direction of what would be expected in serial account), by means of exclusion, we therefore inferred limited parallel processing.